# Terpenoid Transport in Plants: How Far from the Final Picture?

**DOI:** 10.3390/plants12030634

**Published:** 2023-02-01

**Authors:** Olivia Costantina Demurtas, Alessandro Nicolia, Gianfranco Diretto

**Affiliations:** 1Biotechnology and Agro-Industry Division, Biotechnology Laboratory, Casaccia Research Center, ENEA—Italian National Agency for New Technologies, Energy and Sustainable Economic Development, 00123 Rome, Italy; 2Council for Agricultural Research and Economics, Research Centre for Vegetable and Ornamental Crops, via Cavalleggeri 25, 84098 Pontecagnano Faiano, Italy

**Keywords:** terpenoids, carotenoids, apocarotenoids, ABA, strigolactones, crocins, transport, vacuole, apoplast, vascular tissues, specialized metabolites

## Abstract

Contrary to the biosynthetic pathways of many terpenoids, which are well characterized and elucidated, their transport inside subcellular compartments and the secretion of reaction intermediates and final products at the short- (cell-to-cell), medium- (tissue-to-tissue), and long-distance (organ-to-organ) levels are still poorly understood, with some limited exceptions. In this review, we aim to describe the state of the art of the transport of several terpene classes that have important physiological and ecological roles or that represent high-value bioactive molecules. Among the tens of thousands of terpenoids identified in the plant kingdom, only less than 20 have been characterized from the point of view of their transport and localization. Most terpenoids are secreted in the apoplast or stored in the vacuoles by the action of ATP-binding cassette (ABC) transporters. However, little information is available regarding the movement of terpenoid biosynthetic intermediates from plastids and the endoplasmic reticulum to the cytosol. Through a description of the transport mechanisms of cytosol- or plastid-synthesized terpenes, we attempt to provide some hypotheses, suggestions, and general schemes about the trafficking of different substrates, intermediates, and final products, which might help develop novel strategies and approaches to allow for the future identification of terpenoid transporters that are still uncharacterized.

## 1. Introduction

Plant specialized metabolites (PSMs), previously referred as secondary metabolites, are small molecules (<2000 Da) synthesized and accumulated in specific tissues and developmental stages and/or under particular environmental conditions, which own different functions aiming at generally improving plant fitness, including physiological aspects and responses to biotic and abiotic stresses [1]. These metabolites are generally produced in specific organs, cells, or subcellular compartments that are tightly subjected to intra- or inter-cellular transport through biological membranes. Several transport mechanisms have been described and characterized in past years for many classes of PSMs, such as alkaloids, phenylpropanoids, and terpenoids. Among these classes of compounds, terpenoids, also referred as isoprenoids, are the most diverse, with about 80,000 distinct compounds that have important implications in plant growth, development, and adaptation, and in interaction with other organisms [2,3,4]. Relatively few isoprenoids have essential roles in plant development and fitness, such as carotenoids, sterols, gibberellins, chlorophylls, and plastoquinones [2,3,5]. The vast majority of plant terpenoids are PSMs involved in plant–environment interactions [6,7]. For example, diverse terpenoids, including carotenoids, their derivatives, and volatile terpenoids, have indispensable functions in attracting pollinators and seed dispersers [3,8,9], while others have roles in mediating beneficial symbiotic relationships that improve plant fitness [10]. Terpenoids also take part in allelopathic interactions and in abiotic stress responses [11,12]. Many terpenoids are instead potent toxic compounds that serve as chemical defenses against herbivores, insects, and microbial pathogens [2,3,13]. Together with the expansion in terpenoid metabolism, which takes place when land plants face environmental changes and biotic or abiotic stresses [3,5], several mechanisms to improve their differential compartmentation, secretion, and bioactivity have been evolved. In this context, many transporters involved in the secretion of terpenoids have been identified and characterized to date [14,15,16]. In this review, we describe the state of the art of the knowledge about the transport of different types of terpenes, the largest and most diversified class of specialized metabolites produced by plants. Starting from the description of the general transport strategies involved in the translocation of PSMs, we will proceed in detail with the elucidation of the complicate network of movements of different terpenoids inside and outside of cells, including their long-distance through vascular tissues, and their accumulation.

## 2. General Transport Mechanisms for PSMs

The transport of molecules between and within cells generally follows different types of mechanisms, based on the nature of the compound. In more detail, gasses and some small lipophilic compounds, including a few volatiles, can pass through membranes by simple diffusion [17], while charged or bigger molecules, such as metals, ions, sugars, vitamins, volatiles, and general/specialized metabolites, need a protein- or lipid-mediated translocation or vesicle-mediated transport [14,18,19,20,21].

Within cells, the intracellular trafficking of vesicles that bud off from the endoplasmic reticulum (ER) allows the secretion of proteins, lipids, components of the membranes, and other molecules into the plasma membrane and apoplast, or to subcellular compartments, such as vacuole and peroxisomes [18,22,23,24]. The transport is basically obtained by the fusion of vesicles to cellular membranes, with the release of their content. Besides vesicle-mediated transport, the main mechanism responsible for PSMs is protein-mediated transport across membranes [25,26]. The transmembrane transporters generally involved in this process belong to five families: (i) ATP-binding cassette (ABC) transporters, (ii) multidrug and toxin extrusion (MATE) proteins, (iii) nitrate-peptide transporter (NPF), (iv) purine uptake permease (PUP), and (v) AWPM-19 family proteins [14,25,27]. Among them, only ABC transporters mediate the primary active transport (e.g., the transport is driven by the hydrolysis energy of ATP), while the others mediate the secondary active transport (transport driven by an electrochemical gradient). Overall, the different transporter classes are described as follows:ABC transporters function as ATP-driven pumps to import and export molecules of diverse types, such as metals, xenobiotics, hormones, pigments, and largely diversified specialized metabolites [16,28]. The ABC family is highly expanded in the plant kingdom in comparison to other organisms, and in *Arabidopsis thaliana* many members have been functionally characterized [15]. This expansion is mainly due to the fact that plants are sessile organisms that need to adapt to environmental changes and modulate the expression of genes involved in the synthesis, storage, and release of specific metabolites that serve to cope with the modified conditions [29]. In general, nine subfamilies of ABC transporters have been identified in plants (from ABCA to ABCI, with the exception of the ABCH proteins that have not been found yet) [30], with the B, C, and G subfamilies being the most abundant. The members of these subfamilies have been described as being involved in the translocation of PSMs, toxic compounds, and phytohormones [15], molecules that are particularly important for the selective advantage of plants to terrestrial life during evolution [31]. ABCGs and ABCBs are plasma membrane-localized ABC transporters responsible for the translocation of PSMs between cells, while ABCCs are generally present in the tonoplast, where they contribute to the vacuolar accumulation of metabolites. More particularly, the ABCG subfamily, also called pleiotropic drug resistance (PDR) transporters, is one of the predominant class of transporters involved in important biological functions, such as pathogen defense, abiotic stress tolerance, cuticular formation, and phytohormone transport [14,15,32]. For a detailed and updated status of ABC transporter proteins in plants, see [16].MATEs are secondary transporters that use proton or sodium ion gradients to translocate substrates across membranes [33,34]. Like ABC transporters, MATEs have been subjected to a high expansion through tandem and segmental duplication events during the course of evolution. In Arabidopsis, 58 members of MATE transporters, also called DETOXIFICATION (DTX) proteins, have been identified [34]. MATEs exhibit a wide array of functions in plants, including detoxification of xenobiotics and transport of metals, phytohormones, and PSMs [14,25]. Four main classes of MATEs have been identified to date [33]; many MATE members are located in the plasma membrane and function as exporters, but several tonoplast members have also been identified and described as transporters for the vacuolar sequestration of PSMs (especially alkaloids and phenylpropanoids) [21,35,36,37].NPFs, previously known as NRTs, use proton gradient to transport different classes of compounds such as nitrate, amino acids, peptides, phytohormones, and PSMs [25]. After the first discovery of a NPF as a nitrate transporter in *A. thaliana* [38], NPFs were found to translocate amino acids and peptides [39,40]. Lately, it was demonstrated that NPFs are responsible for the transport of glucosinolates and alkaloid derivatives [41,42,43,44,45,46]. To date, eight subfamilies of NPFs have been identified [43] based on the nomenclature proposed in [47].For several years, PUP transporters were thought to be involved in the translocation of purine nucleobase substrates. However, in 2011, it was demonstrated that a *Nicotiana tabacum* PUP-like homolog, named NUP1, is involved in the transport of alkaloid nicotine across the plasma membrane [48]. These findings pave the way for the study of PUP-like proteins as PSM transporters. Two other studies displayed that PUP-like proteins are involved in the transport of a few other alkaloids [49,50]. Interestingly, the PUP-like family originated during terrestrial plant evolution between the bryophytes and the lycophytes. A phylogenetic study showed an extensive pattern of gene duplication and diversification within the angiosperm lineage, with sub-functionalization and neo-functionalization of PUP-like transporters [51].More recently, a new class of transporters, belonging to the AWPM-19 family, has been identified in wheat and rice, albeit they have been also described in Arabidopsis [27,52,53]. AWPM-19 transporter proteins are encoded by an ancient, highly conserved member of the plasma membrane 4 gene family, and they have been found to play a role in abscisic acid (ABA) transport. Actually, in species, such as rice, in which no authentic ABC-type ABA transporters have been identified, the OsPM1 AWPM-19 member has been considered the main transporter responsible for ABA import and drought-related responses [27].

Altogether, of these classes, the most well characterized for the translocation of PSMs are ABC- and MATE-type families.

Besides these classes of transporters, other transmembrane protein families have been described to have a role in hormone translocation and may be discovered in the future to be involved in the transport of other classes of PSMs.

For instance, Sugars Will Eventually be Exported Transporters (SWEETs) are responsible for both the uptake and efflux of sugars in plant cells [54,55,56]. SWEETs are important uniporters for intracellular and intercellular sugar translocation that are strongly induced upon pathogen invasion [57]. The transport is driven by the sugar gradient, and SWEETs localize to different compartments, mainly in the plasma membrane, but also in the tonoplast and the Golgi membranes (see [55] for more details). Surprisingly, SWEET proteins can also be involved in hormone transport (see the description of gibberellin transport in Section 3.3). This discovery has expanded the knowledge on this class of proteins, which can be considered as a broader range of transporters [55].

Another class of transmembrane transporters participating in the transport of phytohormones are PIN-FORMED (PIN) proteins, which are secondary transporters involved in the export of auxins [58,59,60]). The majority of PIN proteins localize to the plasma membrane and transport auxins from the cytosol to the apoplast; other PINs localize to the ER and are responsible for auxin transport from the cytosol to the ER lumen, thus regulating auxin homeostasis [58].

In addition to the translocation mechanisms mentioned above, lipid-transfer proteins (LTPs), which are small proteins of maximum 10 kDa, can play a role in facilitating the diffusion of hydrophobic molecules through membranes or can assist transmembrane transporters, such as ABCGs [61]. In more detail, LTPs can diffuse through membranes and present a tunnel-like hydrophobic cavity, which binds lipids or other hydrophobic compounds that are deposited in the apoplast, such as lignins or suberin [61,62,63]. However, since it has been shown that members of this family are able to bind isoprenoids (e.g., carotenoids) [64,65,66], it cannot be excluded that these proteins might also be involved in the translocation of lipophilic PSMs.

## 3. Function and Transport of Terpenes

### 3.1. Generalities

Terpenes constitute a large and structurally diverse family of primary and specialized metabolites in plants [2,5]. Some terpenes are produced in low amounts and serve as plant phytohormones, such as gibberellins (GAs), cytokinins (CKs), abscisic acid (ABA), and strigolactones (SLs); others, such as carotenoids, are produced in bulk amounts and have a role as light-harvesting pigments and photoprotectors in photosynthesis, or an ecological role as pigments to attract pollinators and seed dispersers. In addition, plants contain a huge variety of monoterpenes, sesquiterpenes, diterpenes, triterpenes, and carotenoid derivatives (apocarotenoids) that function as specialized metabolites with important ecological functions in the interaction of plants with other organisms [6,7,67,68,69]. Terpenes have simple hydrocarbon structures, while terpenoids present different functional groups [2]. Numerous terpenoids are bioactive molecules that are used in medicine (i.e., taxol or artemisinin) or as flavor and fragrance compounds [70,71,72]; therefore, there is a great interest on the elucidation and engineering of the biosynthetic pathways involved in their transport within or between cells.

### 3.2. Biosynthesis and Functions

Terpenes can be classified, based on the number of the five-carbon isoprene units, as hemiterpenes (C5), monoterpenes (C10), sesquiterpenes (C15), diterpenes (C20), sesterpenes (C25), triterpenes (C30), and tetraterpenes (C40). They are all synthesized from the condensation of the five-carbon isoprenoid precursors, isopentenyl diphosphate (IPP) and dimethylallyl diphosphate (DMAPP). IPP and DMAPP are generated by two independent, cell-compartmentalized biosynthesis: the cytosolic mevalonic acid (MVA) pathway, and the plastidial methylerythritol phosphate (MEP) pathway [5]. In the MVA pathway, the precursors derived from acetyl-CoA are condensed by farnesyl diphosphate synthase (FPS) to form farnesyl diphosphate (FPP, C15), which is the precursor of sesquiterpenes, triterpenes, and sterols. On the contrary, in the MEP pathway, IPP and DMAPP are originated from pyruvate and glyceraldehyde-3-phosphate and are condensed by geranyl diphosphate synthase (GPS) to form geranyl diphosphate (GPP, C10), which is the direct precursor of monoterpenes, and by geranylgeranyl diphosphate synthase (GGPPS) to form geranylgeranyl diphosphate (GGPP, C20), which acts as a precursor for diterpenes and carotenoids [5,73].

After the formation of the precursors, FPP, GPP, and GGPP, the different terpenes are generated through the action of terpene synthases (TPS), which are enzymes implicated in the formation of primary terpene skeletons, and are then further modified by the action of various enzyme classes, such as cytochrome P450 hydroxylases, dehydrogenases, and glycosyl- and methyl-transferases [6,73]. These enzymes have various subcellular localization: for instance, P450 enzymes are generally located in the ER membrane [74], while other enzymes are placed essentially in the cytosol or in the ER surface [75,76,77,78]. Differently from terpenes that are synthesized in the cytosol, such as tri- and sesquiterpenes, the biosynthesis of diterpenes and tetraterpenes (and their derivatives) implies the translocation of intermediates from the plastid to the cytosol and/or ER, where the modification enzymes reside. This kind of transport is poorly understood, and several pieces are missing in a puzzle which is still being solved (Figure 1). The final products of terpenoid biosynthesis are then (i) accumulated in the cytosol, (ii) stored in the vacuole, or (iii) secreted into the apoplast. Generally, monoterpenes and sesquiterpenes are secreted through the plasma membrane and accumulate in the apoplast or in specialized structures, such as oil gland or glandular trichomes [79,80]. These molecules have strong ecological roles as phytochemicals against pathogens and herbivore attacks [81,82] or as volatiles emitted for the attraction of pollinators and natural enemies of herbivores [83,84]. Additionally, they are the main constituents of essential oils and are of great interest for their biological properties, such as antioxidant, anticancer, anti-inflammatory, antimicrobial, antiviral, anthelminthic, antinociceptive properties [70,71,72,85,86,87]. Terpenoids also contribute to the formation of complex molecules, such as monoterpene indole alkaloids (MIAs), which are PSMs with powerful pharmacological activities (e.g., the well-known anti-tumor agents vinblastine and vincristine) [88,89]. Among sesquiterpenes, a well-known member is β-caryophyllene, a volatile compound present in many essential oils, especially from clove, rosemary, and *Cannabis sativa,* that plays an important role in the defense against microbial pathogens [90,91]. Triterpenes are the largest subgroup of structurally diverse terpene molecules that includes sterols, steroids, and saponins. Triterpenoids derived from squalene [92] and its members have fundamental roles as wax and resin components, e.g., lupeol and β-amyrin, which are also the precursors of pentacyclic metabolites with distinct bioactivity against plant biotic stressors, such as betulinic acid (from the bark of birch tree, *Betula pubescens*) or glycyrrhizin (found in the roots and stolons of licorice, *Glycyrrhiza glabra*), which are also characterized by their pharmaceutical and, limited to the latter, sweetener properties [93,94]. Diterpenes also have important functions as biotic and abiotic interactions, and some of them act as phytohormones (for example, gibberellins or GAs [78]). Finally, carotenoids are a large group of tetraterpene pigments, with more than 600 different structures [7,9,95]. They are produced in plastids and are the precursors of apocarotenoids, which are enzymatic or non-enzymatic cleavage products that are subjected to several modifications, stored in the vacuoles or secreted outside the cells, and play a role as signal molecules, pigments, aromas, or hormones [96,97,98,99,100].

Lipophilic terpenes can passively diffuse through membranes [101,102,103], or they are transported by sequestration in vesicles or lipid droplets that fuse with the membrane, by carrier proteins, such as LTP proteins, or through specific transmembrane transporters [61,104]. Differently, hydrophilic compounds cannot pass through membranes and are generally stored in the vacuole as conjugated (glucosylated or acylated) compounds [45] by the action of ABC or MATE transporters [45], or through the fusion of ER–derived vesicles [105,106]. The addition of sugars to terpene aglycones occurs later in the biosynthetic pathway, and it represents a mechanism to increase their stability, polarity, and water solubility, and to decrease their biological activity and toxicity [80,107].

In the next two paragraphs, we will describe the transport mechanisms reported, so far, for the main classes of diversified terpenes.

### 3.3. Transport of C10-C15-C20-C30 Terpenoids (Mono-, Sesqui-, Di-, and Triterpenoids)

Although small volatile terpenes emitted by leaves, such as isoprene, are released by simple diffusion across biological membranes [108,109,110], bigger terpenes, including monoterpenes, generally require the presence of active transporters or vesicle-mediated transport [20,111]. For years, it was believed that volatile organic compounds (VOCs) cross membranes by diffusion; however, it was then demonstrated that lipophilic small compounds tend to accumulate into biological bilayers with detrimental effects on membrane integrity, demonstrating that different translocation mechanisms are involved [20,111]; for instance, in *Petunia hybrida*, an ABC transporter is responsible for the translocation of different VOCs across the plasma membrane [111]. Regarding terpenoids, in *Vitis vinifera*, different sesquiterpenoid volatiles are emitted in the flowers, and it has been demonstrated that the sesquiterpene synthase, valencene synthase (VvValCS), which is responsible for their synthesis, is localized to the outer edges of lipid vesicles in pollen grains [112]. Thus, this finding suggests that valencene is stored and secreted through lipid vesicles. Another study provided clues about the involvement of a vesicle-mediated transport of monoterpenes in the secretory cells of the glandular trichomes of *Prostanthera ovalifolia*. Indeed, the electron microscopy images show that plastids (where the biosynthesis of the precursor of monoterpenes takes place) are surrounded by vesicles that then fuse with the plasma membrane [113]. An additional work evidenced that a vesicle-mediated transport from the ER to the plasma membrane is involved in the subcellular movement of the sesquiterpenes copaene and β-caryophyllene in *Sauromatum guttatum* flowers [114]. β-Caryophyllene is a potential toxic volatile since, in the cytosol, it can react with proteins, leading to the formation of caryophyllene oxide (CPO) [115]. CPO induces increased reactive oxygen species (ROS) generation from mitochondria, leading to the induction of degenerative processes [112]. For this reason, the cells need to emit it in the headspace of the plant, and, here, it serves as a defense against bacterial pathogens that invade tissues [90,91]. Interestingly, for this compound, in addition to the vesicle-mediated transport, the involvement of an active transporter belonging to the ABCG subfamily has been demonstrated [81]. Plasma membrane-localized ABCGs are, in fact, the main class of transporters implicated in the secretion of terpenes [15,116]. For example, it has been shown that the exporter AaABCG3/AaPDR3 is responsible for β-caryophyllene secretion in *Artemisia annua* T-shaped trichomes and roots [81]. Of note, this transporter has high homology sequence with *Nicotiana plumbaginifolia* NpPDR1, *N. tabacum* NtPDR1, *Arabidopsis thaliana* AtPDR12, and *Spirodela polyrrhiza* SpTUR2 transporters [81]. The NpPDR1 of *N. plumbaginifolia* transports the antifungal diterpenes, sclareol and sclareolide, across the plasma membrane, resulting in their secretion on the glandular trichomes, where they function as antifungal compounds [117,118]. In contrast, the homolog of this transporter in *N. tabacum*, NtPDR1, is able to translocate not only sclareol, but also the diterpene cembrene [119] and the sesquiterpene capsidiol [120], a phytoalexin involved in the defense against pathogens. Additionally, the NbABCG1 and NbABCG2 transporters of *N. benthamiana* seem to be involved in the same transport [121,122], and an ortholog of the NpABC1 transporter has also been found in Arabidopsis, AtPDR12 [123] and *S. polyrhiza*, SpTUR2 [124]. Notably, SpTUR2 was the first plant PDR transporter to be characterized [125], and its expression in Arabidopsis plants leads to the acquisition of resistance to sclareol [124]. In *P. hybrida*, for instance, it was shown that the PhPDR2 transporter is involved in the defense against herbivores, secreting two steroidal-derived compounds (petuniasterone and petuniolide) [126]. Another well-known example is artemisinin, a potent anti-malarial sesquiterpene lactone that accumulates in the glandular trichomes of *Artemisia annua* [127]. Indeed, Wang and coauthors [128] demonstrated that two transporters from *A. annua*, the AaPDR2 transporter and the lipid-transfer protein 3 (AaLTP3), once expressed in *N. benthamiana* leaves, led to the accumulation of the precursor of artemisinin, (DH)AA (dihydro)artemisinic acid [DHAA]), that is then photochemically converted to artemisinin in the apoplastic space, suggesting the involvement of these transporters in vivo. Finally, a recent study reported the identification of four members of plasma membrane-localized ABCG transporters in *Salvia miltiorrhiza Bunge,* a plant used in traditional Chinese medicine to treat cardiovascular and cerebrovascular diseases; these transporters could be potentially involved in the export of tanshinone (a lipophilic diterpene) and salvianolic acid (a hydrophilic phenolic compound), which are metabolites highly accumulated in the roots and rhizomes of *S. miltiorrhiza* [129]. Additionally, in the present Special Issue, Kato and coauthors described the transport of the triterpene saponin glycyrrhizin in *Glycyrrhiza glabra* (licorice plant) by a H^+^-symporter in the plasma membrane and by an ATP-binding cassette transporter in the vacuole (with a high specificity for the aglycone form) [130].

An additional example of the vacuolar transport of terpenoids is represented by avenacin A-1 and triterpene saponins of oat [131]. Avenacin A-1 is a glucosylated antifungal compound that is stored in the vacuole, and its transport has not yet been elucidated, albeit the co-authors suggested that an ABC transporter might be the main responsible.

Some information is also available for the transport of MIAs. The majority of MIAs are derived from the assembly of tryptamine and the monoterpene secologanin to form the central intermediate strictosidine. The transporter responsible for the vacuolar export of this intermediate was identified in *Catharanthus roseus*: a NPF transporter, named CrNPF2.9, localizes to the tonoplast of leaf epidermal cells and translocates strictosidine from the vacuole to the cytosol [44]. Similarly, a *Solanum lycopersicum* NPF transporter called GORKY is responsible for the export of the steroidal glycoalkaloid α-tomatine and its derivatives from the vacuole to the cytosol [46]. Another characterized transporter involved in MIA translocation belongs to the ABC Family, G group. The plasma membrane CrTPT2 transporter from *C. roseus* is involved in the export of catharanthine to the leaf surface [132].

Regarding diterpene transport, different studies have shed light on the transport mechanisms of the hormone gibberellin (GA). GA biosynthesis starts in plastids, with the formation of the intermediate Ent-kaurene, then proceeds in the ER, and finishes in the cytosol, where different GAs are formed [78]. The mechanisms responsible for the release of the precursors from the plastid and the ER have not yet been identified, while much information regarding the GA movement across the plasma membrane is available. Indeed, GA moves between cells through different mechanisms, with the ion trap being the main mechanism that allows GAs to enter cells [78]. Following entrance into the cytosol, GAs are converted into their charged forms (due to the weak alkaline pH of 7.2), which prevents their further simple diffusion; thus, the export of GAs requires the activity of specific transporters [133]. While GA efflux transporters have not yet been identified, several influx transporters have been described [134]: it was demonstrated in vitro in yeast cells that different NPF transporters are able to transport diverse GAs, as well as ABA (see Section 3.4.1) and other hormones, such as jasmonic acid (JA) [135,136]. In more detail, these studies showed that, of the 45 NPF transporters identified, several are able to import different types of GAs [136,137]. Another type of transporter that has been revealed to be involved in the influx of GAs belongs to the class of SWEET transporters. In particular, the *A. thaliana* AtSWEET13 and AtSWEET14 transporters have been shown to be able to transport GAs in different forms (active and non-active) [138]. In addition, in rice, a SWEET transporter, OsSWEET3a, was found to be involved in the transport of GAs [139]. Interestingly, GA transport by SWEET proteins evolved independently during plant evolution: in fact, in Arabidopsis, it arose from sucrose SWEET transporters, while in rice, it arose from glucose SWEETs [139]. GAs are also accumulated in the vacuole, especially as a reservoir to use during the suberization process. It was demonstrated that GAs, as well as ABA, are loaded into the pericycle vacuoles at the phloem in the roots of *A. thaliana* to form a storage pool, which is later released into the endodermis to induce suberization [140]. An NPF transporter, named NPF2.14, is a tonoplast-localized transporter in pericycle cells able to transport GAs and ABA (see also Section 3.4.1) from the cytosol to the vacuole. When the root elongates and GAs and ABA levels in the vacuole become high, the hormones are exported out of the pericycle vacuole and imported into the endodermis by the NPF3.1 transporter [140].

### 3.4. Transport of C40 Terpenoids (Carotenoids) and Derivatives (Apocarotenoids)

As previously stated, one of the major missing pieces of the complex puzzle of terpene transport is with regard to the transport of carotenoid intermediates. Carotenoids accumulate in plastids [141] or on their surface (Figure 1), where they are then cleaved by carotenoid cleavage dioxygenase (CCD) enzymes or by spontaneous non-enzymatic mechanisms (ROS) [142], generating intermediates that are subsequently subjected to modifications in the cytosol and/or in the ER to produce active apocarotenoids [96,97,98].

So far, the translocation of these intermediates is poorly studied. In 2014, it was described a mechanism, based on hemifused-membranes between plastids and the ER, that allows enzymes to interact with non-polar compounds in both organelle membranes [143]. Similar contact points were described for the ER with other organelles, including the vacuole [144], suggesting that a possible way of translocation of the intermediates and the final apocarotenoids products can involve physical contact between subcellular compartments. However, the existence of specific transporters able to translocate carotenoid metabolites cannot be excluded. In fact, in mammalian cells, it has been recently discovered a class of proteins involved in the translocation of carotenoids [65]. In more detail, the so-called ASTER proteins create a bridge between the plasma membrane and the ER, promoting the translocation of cholesterol among these compartments. It has been demonstrated that ASTER proteins are also able to bind carotenoids [65]. Members of ASTER-like proteins have not been described in plant, but, recently, a chloroplast protein (named CPSFL1) was reported to bind and transport carotenoids in *Chlamydomonas reinhardtii* [145]. The CPSFL1 protein is characterized by two CRAL-TRIO domains, which are structurally conserved domains found exclusively in eukaryotes and consist of alternated α-helices and β-strands that can bind different small hydrophobic molecules, such as α-tocopherol, retinaldehyde, squalene, and phosphatidylinositol. Thus, this study opens the way to future research on the identification of translocators for carotenoid metabolites in higher plants.

Differently from carotenoids, much evidence starts to be available for apocarotenoids that have phytohormone activity, such as ABA and SLs, or for pigments, such as crocins. In the next paragraphs, we will provide a description of the transport processes of these apocarotenoids by detailing their biosynthesis and presenting a series of findings, which can be helpful to clarify the transport of molecules with similar biosynthetic pathway compartmentalization.

#### 3.4.1. Transport of ABA

ABA is a well-known hormone involved in a series of physiological processes and responses, such as stomata closure, embryo maturation, fruit ripening, cell senescence, and abiotic/biotic stresses [146,147,148,149]. From a chemical point of view, ABA is an isoprenoid synthesized in the frame of carotenoid catabolism. To date, two pathways have been described. The first pathway is the canonical one, in which two plastid-localized xanthophylls, 9-*cis*-violaxanthin and 9-*cis*-neoxanthin, act as the initial substrates for a specific class of CCD (carotenoid cleavage dioxygenase) enzymes named NCED (9-cis-epoxycarotenoid cleavage dioxygenase) to give rise to the intermediate xanthoxin that is translocated from the plastid to the cytosol and then converted into ABA-aldehyde, which is finally converted to ABA [150,151]. The second pathway is a more recently identified but not yet elucidated route, starting with β-carotene and/or zeaxanthin, and involving a group of β-11-apocarotenes (β-11-apocarotenal and 3-OH-β-11-apocarotenal in *cis*-/*trans*-isomeric forms) as xanthoxin precursors [152]. ABA is finally converted into a group of catabolites, including ABA-glucose ester (ABA-GE), 7′-/8′/9′-hydroxy-ABA (OH-ABA), and phaseic, dihydrophaseic, and neophaseic acids (PA, DPA, and NPA, respectively), which are stored in the vacuoles and in the apoplast [153].

ABA virtually represents the main form with hormonal properties, albeit additional hormonal and hormone-like bioactivities have been reported for PA and OH-ABA, whereas ABA-GE is the only catabolite that can be reversibly reconverted to free, active ABA following a glucosidase (BG2)-catalyzed reaction in a mechanism used by plant cells to obtain an immediately available source when they are exposed to stress effectors [154,155,156].

The primary sites of ABA synthesis are the vascular tissues, and it is subsequently subjected to transport at the subcellular, cell-to-cell, tissue, and organ levels in order to exert a vast range of physiological processes [157,158]. Basically, ABA is found in two forms in plants: anionic (ABA^−^) and protonated (ABAH), with the latter being able to rapidly diffuse across cell membranes (passive transport [159]). However, ABA can also be actively transported by transporters, which have been initially identified in rice and Arabidopsis [160,161] and can be classified into the following families: (i) ABC, (ii) NPF, (iii) MATE, and (iv) AWPM-19 family proteins (OsPM1) [162]. Overall, short- and long-distance transport of ABA is mediated by ABC transporters, which belong to different classes according to their specificity. In Arabidopsis, 129 ABC transporters involved in ABA have been reported so far, thus suggesting a great extent of functional redundancy [161]. More specifically, sub-classes C and G are the ones reported, thus far, to participate in ABA transport, with specific roles inside the cell and from inside to outside the cell (and vice versa), respectively.

At the subcellular level, ABA-GE is moved from the cytosol to the vacuoles thanks to two ABC transporters, ABCC1 and ABCC2, which were firstly isolated in Arabidopsis and have been characterized by a divergent substrate affinity, with the latter member being the one with the highest transport capacity [99]. Notably, neither *atabcc1* and *atabcc2* single mutants or *atabcc1 atabcc2* double mutants display ABA-related phenotypes, thus suggesting the presence of multi-specific transporters of other classes (e.g., MATE), which might be involved in ABA-GE vacuolar sequestration [99]. On the contrary, several ABCG members are involved in ABA medium- and long-distance transportation in strong association with specific physiological processes. For instance, in Arabidopsis, the AtABCG40 and AtABCG22 transporters mediate ABA uptake in plant cells, thus controlling stomatal closure [163], whereas the plasma membrane-localized AtABCG25 exports ABA from vascular tissue to several plant parts [164], and its overexpression results in an enhanced stomata response, providing evidence of an active transport to the guard cells [165]. A similar function has been described for AtABCG17 and 18, with a further role on ABA homeostasis balance, either in normal and abiotic stress conditions, which results in a rigid control of its translocation from the shoot to the root in the process of lateral root emergence [166]. Interestingly, AtABCG40, together with AtABCG30, has also been shown to regulate seed germination by ABA import; similarly, AtABCG25, AtABCG31, and NRT1.2/NPF4.6 (see below) can control the export of ABA from the endosperm to different plant parts and from xylem/phloem to guard cells, respectively, thus demonstrating (i) a high promiscuity of a single ABC in participating in multiple events of plant physiology and that (ii) several ABC members take part in the same process [161].

A role for ABA transport has also been reported for other classes of plant transporters. Indeed, a DTX/MATE-type transporter named AtDTX50 has been found to be active in ABA efflux in vascular tissues as well as in guard cells [167]. Of interest, the defective *dtx50* mutant was characterized by retarded growth, likely due to high ABA sensitivity/endogenous contents [158]. A great advance in the elucidation of ABA transport-derived effects has been achieved by Qin et al. [168], who revealed, in rice, the molecular basis underlying seed development that occurs with the involvement of ABA produced at the leaf level and transported by the action of defective grain-filling 1 (DG1), a MATE transporter that effluxes ABA at nodes and rachilla and which activity is temperature-dependent [168]. Similarly, a group of NRT/NPF transporters have been found to play fundamental roles in ABA-derived physiological responses. The first identified member acting as an ABA importer was NRT1.2/NPF4.6, whose mutant and overexpression lines displayed, respectively, reduced and higher ABA sensitivity during seed germination and vegetative growth; in addition, the latter was also characterized by a lower surface temperature of the inflorescence stems by virtue of the excess water loss from the open stomata [135,169]. However, it has been subsequently shown that NRT1.2/NPF4.6 stability and ABA import activity are finely regulated in a negative fashion through a direct interaction with the C-terminally encoded peptide receptor 2 (CEPR2), which influences its phosphorylation and degradation in response to environmental stress [170]. More recently, an additional notable member named NPF5.1 was found, which has been reported to import ABA, thus influencing stomatal movement and seed germination [169,171], while ABA transport capacity has also been confirmed for NPF4.2, 4.5 and 4.7 [172]. Finally, a new class of tonoplast-localized NPF-ABA transporters, named NPF2.14, was identified, which is able to accumulate in the epidermis during root elongation and regulate its suberization. Moreover, additional members (NPF2.12 and 2.13) present at the plasma membrane level and acting as ABA importers were also discovered [140]. Notably, the identification of these specialized transporters proves the existence of cell type-specific and long-distance transport of ABA, with important implications about its role in the regulation of plant physiology. A last class of ABA transporters is represented by the AWPM-19 family proteins, which have been identified and characterized in rice (OsPM1) [27] and Arabidopsis [53], and which appear to act as osmosensors for ABA influx. These proteins are located at the plasma membrane level, and their expression is regulated by the AREB/ABF family transcription factor OsbZIP46. Overall, these proteins guarantee a prompt plant response to drought and other abiotic stressors, and they participate in crucial plant physiological processes, such as embryo maturation, to trigger seed germination. An interesting case is the one represented by the investigation of the Lr34 (leaf rust 34) locus from wheat (*Triticum aestivum*), which confers aspecific resistance against numerous fungal diseases and is one of the most frequently used resistance genes in wheat breeding worldwide [173]. In more detail, Krattinger et al. [174] identified the molecular source of the Lr34 locus, which corresponds to a full-size ABCG transporter modulating (i) ABA and phenylpropanoid local concentration in the leaf, (ii) the remodeling of phospholipid (PL) profiles in the plasma membranes, and (iii) the promotion of the constitutive expression of a series of stress-upregulated genes [174,175,176]. Interestingly, Lr34 can occur either as susceptible (Lr34sus) or resistant (Lr34res) allele variants, with only Lr34res being detected in planta, thus suggesting the existence of post-transcriptional or post-translational mechanisms controlling Lr34 accumulation [176].

#### 3.4.2. Transport of SLs

The discovery of SLs began in the 1960s with the first report of plant root exudates triggering the germination of the parasitic plant *Striga*. Since then, enormous progress has been achieved in both understanding the biosynthesis and functions of these apocarotenoids.

SL biosynthesis is a process that has been biochemically investigated in several species, including model plants and crops (e.g., Arabidopsis, pea, and rice). A sequential plastid-localized reaction from all-*trans*-β-carotene via 9-*cis*-β-carotene and 9-*cis*-β-apo-10′-carotenal, which is catalyzed by a carotenoid isomerase (D27) and two distinct CCDs (CCD7, CCD8; names vary in different species), respectively, produces carlactone (CL), the common precursor of SLs. The cytosolic enzyme MAX1, belonging to the family of CYP711, catalyzes the oxidation of CL into carlactonic acid (CLA), which is further oxidized by the species-specific cytochrome P450 to produce the so-called canonical SLs (e.g., orobanchol and solanacol) and/or non-canonical SLs (e.g., heliolactone and zealactone) [8,177,178].

Following their discovery, SLs have been chronologically reported as inducers of hyphal branching in symbiotic arbuscular mycorrhizal fungi (AMF), as regulators of plant shoot lateral branching, and as phytohormones contributing to shaping the architecture of the plant, with novel functions adding on a yearly basis [179].

Such complex processes require the following: (a) a mechanism of control of SL biosynthesis; (b) the mobilization of SLs, as signal, both within the plants (i.e., long-distance signaling) and in the soil (i.e., short-distance signaling); and (c) a signal transduction mechanism. The points (a) and (c) will not be discussed in this work but have been reviewed elsewhere (see [177,178]).

Plant roots contain relatively high levels of SLs compared to other tissues, and several grafting experiments using a range of combinations of *d27*, *ccd7*, *ccd8*, and *max1* mutants have demonstrated that (a) SLs synthetized in roots can move acropetally; (b) a basal level of biosynthesis is also present in the above-ground tissues; and (c) CL, CLA, and/or its metabolites methyl carlactonoate (MeCLA) and hydroxyl-methyl-carlactonoate (1′-OH-MeCLA) could act as the long-distance signaling molecule/s. Additional studies are necessary to shed light on the nature of the transported molecule/s and if a xylem or a cell-to-cell movement is involved [178].

Concerning short-distance signaling, SLs are released into the rhizosphere, providing early plant–AMF interaction signal [10] and germination stimulus for parasitic plant seeds (reviewed in [180]).

SL exudation appears to be an active process in *Solanaceae*, relying on the G-type ABC transporter orthologues of *P. hybrida PhPDR1* (PLEIOTROPIC DRUG RESISTANCE 1) gene, which was first identified. This transporter was shown to be expressed in the shoot lateral axils, close to the lateral buds, in the root tips, and in the outer lateral membrane of specific non-suberized exodermis cells named hypodermal passage cells (HPCs), that function as entry points for AMF. Moreover, the expression levels were augmented by phosphorus starvation (Pi-) and the application of exogenous GR24, a synthetic SL analogue. Interestingly, the petunia mutant *pdr1* showed a reduced amount of orobanchol in root exudates and a high branching phenotype, even if its levels in root extracts was similar to the wild-type ones [181]. Therefore, a model was proposed where the ABCG transporter PhPDR1 acts by (a) unloading SLs from biosynthetic tissues (e.g., the root tips), (b) exuding SLs to the rhizosphere; and (c) loading SLs into dormant buds. Notably, the *pdr1* mutant rootstock can complement the *ccd8* mutant, thus suggesting the existence of a PDR1-independent transport route (e.g., other transporter) or substrates [182].

Additionally, the overexpression of *PhPDR1* in petunia further confirmed the SL transport activity of this gene. Notably, an increment of total biomass and a reduced leaf senescence phenotype were also observed in the overexpression lines [183]. Of interest, the *N. tabacum NtPDR6* has been characterized as a possible orthologue of *PhPDR1*, with an inducible expression by inorganic phosphate (Pi) and an increase in branching in RNAi lines. However, SLs have not been assessed in this work [184].

Finally, in *Solanum lycopersicum*, the genes *ABCG44* and *ABCG45* have been knocked-out by the CRISPR/Cas9 technology, and edited lines have been characterized. Overall, both the single mutant *abcg44* and the double mutant *abcg44*/*abcg45* showed (i) increased branching and reduced orobanchol content in root exudates, at an even higher extent in the double mutant, and (ii) decreased susceptibility to the parasitic plant *Phelipanche aegyptiaca* [185]. However, the role of *ABCG44* and *ABCG45* on SL transport still needs additional clarifications.

Interestingly, although *Fabaceae* exhibit exodermis-free roots and, consequently, they do not have specialized HPCs, the gene *MtABCG59* in *Medicago truncatula* resulted in an orthologue of *PhPDR1*, with a wider pattern of expression in root compared to petunia. Of note, the root exudates of the mutant line *mtabcg59* (retrotransposon insertion) induced a lower germination rate with seeds of the parasitic plant *Phelipanche ramosa*, thus providing an indirect evidence of SL reduction in root exudates. However, a passive diffusion mechanism of SLs in the rhizosphere was also hypothesized [15,186].

#### 3.4.3. Transport of Crocins

Crocins are glycosylated apocarotenoid pigments that originate from the oxidative cleavage of carotenoids, such as zeaxanthin, β-carotene, and lycopene. The main source of crocins is saffron, the dried stigmas of *Crocus sativus*, in which crocins originate in the stigmas from the cleavage of zeaxanthin by the CsCCD2 enzyme [187] and undergo subsequent dehydrogenation, glycosylation, and storage in the vacuole, where they can reach up to 10% of dried matter [77,100,188,189,190]. Crocins are also accumulated in the fruits of *Gardenia jasminoides*, where their synthesis starts from the carotenoids β-carotene, zeaxanthin, and lycopene by the action of GjCCD4a [191], and in the flowers of *Buddleja davidii*, in which the substrate is zeaxanthin and the enzymes involved are BdCCD4.1/BdCCD4.3 [192]; moreover, for these species, the whole pathway from the carotenoid precursor to the generation of crocins with different degree of sugar moieties have been fully revealed [191,193].

In *C. sativus*, the crocin biosynthetic pathway involves different compartments: the synthesis, indeed, starts in the chromoplast with the cleavage of zeaxanthin by the plastid-localized CsCCD2, leading to the formation of the C20 compound crocetin dialdehyde [77,194], continues in the ER with the dehydrogenation of crocetin dialdehyde into crocetin through the action of the ER-localized CsALDH3I1 enzyme [77], and ends in the cytosol with the action of the two steps of glucosylation, which is performed by CsUGT74AD1 [77] and CsUGT91P3 [188], although an additional UGT member named UGTCs4 has been recently reported [190]. Finally, crocins are transported in the vacuole, where they are stored at high concentration. One tonoplast-localized *C. sativus* ABC transporter, CsABCC4a, seems to be capable of transporting crocins with different degrees of glycosylation and stereoisomerism across membranes [100]. This transport is specific for ABC transporters (since CsABCC2 has also been shown to translocate crocins in in vitro transport assays), while the MATE transporters tested (CsMATE1a, CsMATE1b, and CsMATE4) display a specific transport activity only for flavonoids. Of interest, another apocarotenoid named picrocrocin, which is produced by the subsequent action of CsCCD2 [194], and a glycosyltransferase named UGT709G1 [195] were not found to be transported to the vacuole [100], thus suggesting that either (i) they reside in the cytosol or (ii) another still unidentified transporter is playing this function. Thus, through a transportomic assay [196], it is possible to test the transport of several metabolites all together by using total raw metabolite extracts, and this approach has revealed that the transport of crocins takes place cooperatively, at least for crocins with three glucose moieties (crocin 3): indeed, the transport of the molecule in *trans* configuration occurs only at high concentration and in the presence of the *cis* isomer [100]. This finding suggests that, in vivo, the transport of certain substrates can be influenced by the presence of other molecules. This aspect needs further investigation but paves the way for future transport assay screening. Notably, the expression of CsABCC4a in combination with CsCCD2 in *N. benthamiana* leaves led to crocin accumulation at higher levels compared to that of CsCCD2 alone [100]. This indicates that the expression of the crocin tonoplast transporter can enhance crocin accumulation, probably due to positive feedback mechanisms. This strategy can be applied to other transporters/metabolites in a metabolic engineering strategy to improve the yield of the desired products. Another aspect of interest regards the activity of heterologous transporters able to transport crocins: Martì et al. [197] used a virus-driven system to transiently transform *N. benthamiana* leaves with CsCCD2 and showed that crocins accumulated in the vacuoles, thus demonstrating the existence of endogenous transporters that are able to translocate these compounds in the transformed leaf cells.

Crocins are stored in the vacuoles of stigmas until the senescence of the plant leads to the mechanisms of recycling/reusing [198]. Interestingly, it has been demonstrated that crocins are transported from the senescent stigmas to the ovaries and then to the new developing corms. Once arrived in the corms, crocins are deglucosylated to crocetin, and the released glucose moieties are used during corm development. This evidence sheds light on the involvement of different short- and long-distance crocin transporters and/or transport mechanisms from the vacuole to the cytosol, then to apoplast, and finally through the xylem to the final destination in the corms. However, the compartment in which crocetin accumulates in corms has not been characterized and the putative transporters not yet identified.

An interesting paper that can help the discovery of apocarotenoid transporters in saffron is the one by Mohiuddin et al. [199], in which 77 ABC transporter genes from *Crocus* transcriptome were identified and classified into eight subfamilies based on their similarity with *A. thaliana* transporters. Among them, ABCB and ABCG were the most predominant, although they were mostly expressed in anthers and tepals. Differently, the ABCC, ABCD, ABCE, and ABCF members were expressed mainly in stigma, suggesting a possible role in the transport of saffron apocarotenoids. In fact, one year later, it was demonstrated that CsABCC4a and CsABCC2 transporters were involved in the vacuolar accumulation of crocins [100]. Mohiuddin et al. in 2018 also proved that *ABCB* and *ABCC* genes showed significant induction in response to hormones, including the diterpene gibberellic acid and the apocarotenoid ABA for ABCB members, and ABA for ABCC, thus providing clues about their possible role in the transport of these hormones. Additionally, they identified a few orthologs of the AtABCG40 transporter that took part in the transport of SLs (see Section 3.4.2) [181], suggesting that they might be redundantly involved in similar functions. Since ABCG members are generally implicated in the transport of specialized metabolites across the plasma membrane [16], the identified *C. sativus* ABCG member could also have a role in the long-distance transport of crocins.

From the synthesis of the precursors (green dots) through the cytosolic MVA (*a’*) and plastidial MEP (*a*) pathways, different terpenes (*b’* and *b*) are formed in the cytosol or plastid, respectively. These terpenes are then modified into terpenoids (*c* and *d*) by different enzymes that localize to the ER and/or cytosol. The translocation of the intermediates from the plastid to these compartments has not yet elucidated, but could involve mechanisms based on hemifused membranes between plastids and the ER [143]. The final products of the terpenoid biosynthetic pathways are then stored in the vacuoles (*e*), secreted into the apoplastic space (*f*), or imported inside the cells through the action of transmembrane transporters. To date, only members of the ABC, MATE, NPF, AWPM-19, and SWEET families have been reported to be involved in the translocation of terpenoids (indicated in the figure). A sixth class consisting of PUP transporters might also contribute, since they are responsible for the transport of other PSM-chemical types (e.g., nicotine) in plant cells. Picture created with BioRender.com.

## 4. Discussion and Future Directions

Although the biosynthetic pathways of different classes of terpenoids have been well studied for a long time, and, in many cases, the enzymes responsible for their synthesis and catabolism have been identified and characterized, the subcellular compartmentation and the long-distance transport of these molecules are generally poorly elucidated, with some exception (such as the vacuolar storage of crocins and glucosyl esters of ABA, or the transport of ABA through the vascular tissues). More is known about the secretion of terpenoids: indeed, many lipophilic compounds with roles in plant defenses are secreted through the plasma membrane and accumulated in the apoplast or in specialized structures, such as glandular trichomes. The preeminent class of transporters involved in the secretion of terpenoids with ecological roles is that of the ABC, subfamily G, transporters [14,15,16]. Indeed, duplication events occurred in line with chemodiversity during the evolution of Angiosperms, leading to the expansion of the ABCG transporters involved in the secretion of compounds of defenses or involved in ecological relationship [186].

Increasing the knowledge about ABC transporters, as well as on other classes of PSM transporters, could be useful to generate new idiotypes through the approaches of genome editing and/or metabolic engineering to modify the sequence of transporters and, thus, to alter their substrate specificity. In the perspective of a sustainable agriculture, these approaches might be of great efficacy, especially for ABC transporters involved in the secretion of compounds with symbiotic interaction activities, such as SLs [16] or the recently identified terpenoid zaxinone [200] (see below). The goal of sustainable agriculture is, in fact, to create new varieties with increased productivity and low impact on the ecosystems (e.g., less pesticides and less fertilizers) in a strategy defined by “zero waste”. In this context, ABC transporters would be ideal targets for several desirable traits to improve yield (either in terms of total biomass and productivity) as well as crop quality [15].

One of the more elusive bottlenecks of the identification of transport proteins is related to their functional characterization. In this respect, different approaches have been used to date, but a series of considerations has to be remarked: (i) in vivo approaches, such as silencing, editing, and overexpression, need the availability of genetic information and genetic tools; (ii) the expression in heterologous systems, such as yeast cells, Xenopus oocytes, or insect cells, and in vitro assays often present technical difficulties as gene cloning (transporters are usually big in size and present many hydrophobic transmembrane domains); and (iii) a huge diversity of substrates needs to be tested; however, in order to overcome this aspect, transportomic approaches have been developed, consisting of the use of total metabolic extracts, which allows for the simultaneous testing of hundreds of largely diversified compounds [196]. These approaches support the identification of novel substrates and activities, although the phenomena of inhibition and competition can occur, making their characterization even more complicated.

Omics technologies (genomics, transcriptomics, proteomics, metabolomics, and fluxomics) are powerful tools to be exploited for the characterization of PSM transporters [116]. Recently, the One Thousand Plant Transcriptomes (1KP) initiative was launched [201,202], which collects transcriptomic data from more than 1000 species and might potentially enable the capture of novel transporters by virtue of the large dataset generated. In addition, studies of particular utility are those in which transcriptomic analyses are performed on plants with altered expression of biosynthetic genes (by silencing and overexpression), in order to identify up- and down-regulated genes tentatively involved in their catabolism and transport (see for example [203]). Finally, integrated omics analyses, including genome-wide association studies (GWAS) and gene co-expression networks (GCNs), represent important strategies for the discovery of new gene functions, and many studies are now focused on the GWAS and GCNs of PSM transporters [100,204].

The present review reveals that, of the thousands of terpenoids identified in nature, only a few of them have been characterized from the point of view of their transport activity (Figure 1). Besides the ABC family, which represents the main class of transmembrane transporters involved in the sequestration of terpenoids into the vacuole, on their secretion out of the plasma membrane, and on their import inside the cell, other classes of translocators have also been described. Members of MATE, NPF, and AWPM-19 transporters are involved in the ABA movements inside and outside of cells. An interesting observation is that MATEs are transporters involved in the transport across tonoplast or plasma membrane of a large number of PSMs, in particular alkaloids and phenylpropanoids [14,18,205], but they seem poorly involved in the translocation of terpenoids. NPFs have a well-defined role in the import of ABA and GAs across plasma membranes and on their uptake into the vacuoles [135,136,137,140,169,170,171,172]. NPFs are also involved in the transport of MIAs and steroidal alkaloids, being responsible for the vacuolar export of the strictosidine in *C. roseus* [44] and of α-tomatine in *S. lycopersicum* [46].

These findings suggest a possible involvement in the transport of other PSMs, including other terpenoids. Another group of transmembrane transporters that deserve a better extent of investigation as candidate terpenoid transporters are (i) PUPs, which have been reported, to date, to transport alkaloids [48,49,50], (ii) SWEETs, and (iii) AWPM-19 proteins, which import GAs ([138,139]) and ABA ([27,53]), respectively.

The general scheme for the accumulation of hydrophilic terpenoid compounds in the vacuoles has been postulated. In more detail, the precursors are synthesized in plastids, the intermediates are localized in the ER and/or in the cytosol, and the final products, which are glycosylated, acetylated, or have other modifications, are stored into the vacuoles by the action of a tonoplast ABC transporter. Such scheme has been described for ABA-GE [99,159] and crocins [77,100], and can be also applied to the accumulation of diterpenoid steviosides [206] or of crocins in *G. jasminoides* and *B. davidii*, species that accumulate these compounds and for which the biosynthetic genes have been identified [191,192,193], although no information is known regarding their transport and storage. In general, ABA is the most well-studied terpenoid in terms of its transport at the subcellular [99], cell-to-cell [165], long-distance (by vascular tissues) [164], and, more recently, cell-type levels [140].

On the contrary, mechanisms underlying the transport and accumulation of a series of apocarotenoids still need to be fully depicted, and this aspect is of particular interest for molecules acting as signal molecules in ecological interactions, such as β-cyclocitral, anchorene, zaxinone, and mycorradicin [10,97], or new pigments, such as azafrin. The latter compound is released from the root cells of *Escobedia grandiflora* and accumulates in the apoplast; in this way, azafrin can provide protection from oxidative damage caused by biotic or abiotic stresses [207], while having a potential industrial exploitation as novel colorant. In this context, a series of clues might come from other well-known apocarotenoids with signal and hormonal activities, such as SLs, which share several characteristics (at chemical, pathway, and subcellular localization and functional level) with the aforementioned compounds. This might suggest the occurrence of similar transport processes in the already known transporters, although the existence of novel, uncharacterized members cannot be excluded. Another intriguing point to be taken into consideration is related to the synergic and antagonistic effects occurring between terpenoids with hormonal and hormone-like activities, for example, GAs and ABA, which antagonistically influence various plant developmental processes and environmental responses (reviewed in [208]), or ABA and SLs, for which an interaction between the two pathways has been reported [209,210,211], and, finally, the apocarotenoid zaxinone, which negatively regulates SL biosynthesis in rice [200] and positively in Arabidopsis, along with ABA [212]. All of these findings, together with the occurrence of (i) positive and negative feedback regulation events [213,214], (ii) dose-dependent effects [215,216], and (iii) cross-links between different terpenoids (reviewed in [217,218]), must have a strong impact on the modulation of their transport (besides the controls on transcription and translation), which has to be tightly and finely coordinated according to the time/stage in which the required activity has to be exerted.

An additional issue to be addressed is related to the translocation of plastid-localized intermediates of terpenoids that can involve plastidial transmembrane transporters or the interaction between organelles membranes [143]. In this perspective, the secretion of the apocarotenoids or GA intermediates outside of plastids is still completely undiscovered and will need to be sorted out. An exemplificative question is: how xanthoxin (ABA precursor), carlactone (SL precursor), crocetin dialdehyde (crocin precursor), or ent-kaurene (GA precursor) exit out of plastids to give rise to their final products?

A last fascinating food for thought is the recent discovery of the so-called ASTER proteins. In mammalian cells, these proteins have been demonstrated to have a role in carotenoid transport [65], in addition to an ABC transporter (ABCA1), which is also hypothesized to be involved [219], thus suggesting that transporters participating in the translocation of lipids should be better investigated. To date, members of ASTER-like proteins have not been described in plants, but a plastid protein, CPSFL1, was reported to bind and transport carotenoids in *C. reinhardtii* [145]. This protein has characteristic domains that make it able to bind different small hydrophobic molecules, including terpenoids, and might represent a good candidate for the translocation of intermediates from plastids to other compartments. Thus, these studies open the way for future research on the identification of translocators of carotenoid metabolites in higher plants. In this respect, the interactions between the plastidial outer envelope and the endomembrane system are thought to be involved in the trafficking of molecules (especially hydrophobic ones) between plastids and the ER [143,220]. In addition, PIN proteins might potentially play an important role in terpenoid transport: in fact, they are mainly localized to the plasma membranes, where they are involved in the secretion of auxins, but they have also been found on ER surface, regulating the movements of auxins from the cytosol to the ER lumen [58].

In conclusion, although several advances have been achieved on the research area of terpenoid trafficking, many fundamental questions have to be addressed, and we are still far from the final picture. The full elucidation of the transport mechanism of terpenoids are fundamental to finely tune the optimal metabolic engineering strategies for the production of these high-value bioactive compounds in heterologous systems, for their industrial production and exploitation, and for the implementation of the beneficial interactions between plants and useful microorganisms (e.g., mycorrhizae) to finely regulate plant growth in innovative agricultural contexts.

## Figures and Tables

**Figure 1 plants-12-00634-f001:**
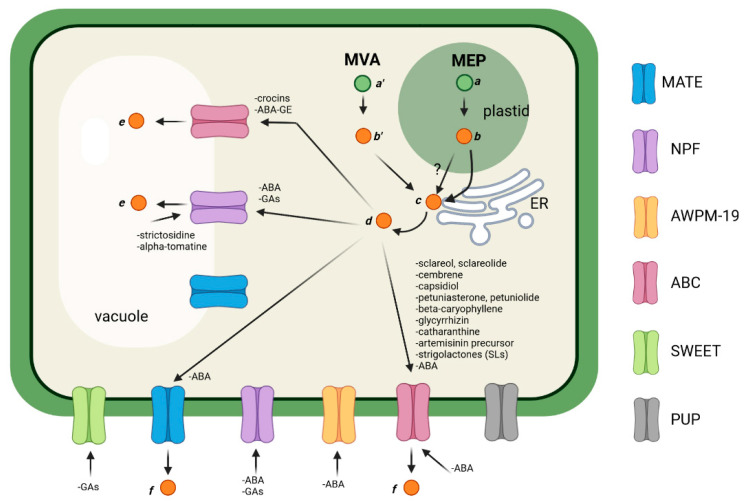
General scheme depicting the transport of terpenoids in plant cells.

## Data Availability

Not applicable.

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
