# Peer review of "Terpenoid Transport in Plants: How Far from the Final Picture?"

_plants, 2023, doi:10.3390/plants12030634_

Round 1

Reviewer 1 Report

This review article on the terpenoid transport in plants is very complete, has very up-to-date information, and reflects very well where research on this transport is currently focused and the most important gaps that exist. I like the writing of the manuscript, easy to understand and at the same time very well documented. The distribution of the different sections helps to read the text, it makes reading the manuscript easy and enjoyable. In my opinion, the manuscript can be published with some minor modification, which is listed below, along with some questions.

I would like if it is possible for figure 1 to be expanded a bit, including some non-terpenoid compounds such as alkaloids.

I also believe that other important activities that have been described for essential oils should be incorporated into the text, in addition to those written, such as antimicrobial or anxiolytic activities, for which some complementary reference should be added. Thus, the identical activities would not be repeated on lines 210 and 213, because I think it is repeating information.

I would like you to clarify why cells need to emit caryophyllene in the headspace of the plant, line 256. Emit it to attract pollinating insects?

As a curiosity, is anything known about transporters that help the accumulation of sesquiterpenes at the level of the essential oil vesicles of the citrus flavedo?

Regarding stylistic errors, I have detected small mistakes in the words mechansims, sequiterpenes and form, in lines 149, 198, and 200, respectively. I also think that in line 262 Nicotiana tabacum should appear as N. tabacum and in line 289 H+ as H+.

As for the reference of line 288, I think Terasaka should be replaced by Kato, although I don't know if Plants allows citation with the name of the last author, instead of the first.

In line 359, wouldn't it be better to write "are finally converted to ABA" than "is finally converted to ABA"?

In the final references, some doi are missing (references 3, 8, 43, 49, 50, 55, 65, 73, 77, 82, 87, 88, 94-101, 103-106, 108-109, 124, 142, 143, 160-165, 167, 169-172, 184, 190 and 192). Moreover, for some journal and article titles all the initials of the words are written in capital letters, and sometimes not, and this should always be written the same.

Some species do not appear in italics on article titles, such as Artemisia annuaNicotiana plumbaginifoliaA. annuaArabidopsis thalianaClamydomonasA. thalianaB. napusMedicago truncatulaCrocus sativus, Buddkeja davidiiEscobedia grandiflora (lines 919, 991, 1016, 1036, 1058, 1066, 1085, 1165, 1174, 1182, 1224, respectively).

The journal Recent Res Dev Cell Biol should not be written with abbreviation (line 953), neither Plant Cell Rep(line 1051), Plant Signal Behav (line 1122), Int J Mol Sci (line 1172), J Exp Bot (line 1182 and 1231) or Int J Mol Sci (line 1252).

Why is the 155 reference not cited as the 175 reference -available at Research Square-?

In lines 148-149 is necessary to write “an international journal of experimental plant biology”? (reference 148)

Is citation 175 missing authors?

Author Response

Dear reviewer,

thank you for your suggestions and comments. Please find our reply in attachment.

Best regards,

Olivia Demurtas

Reviewer 2 Report

The review entitled, Terpenoid transport in plants: how far from the final picture? is concise well written and timely.

Abstract- well structured with clear background, research gap and objectives.

Introduction- Looks short. The significance of terpenoids in plant life can be elaborated.

Other subtopics are written well and no changes are required.

Authors have pin pointed the exact research gap and given future directions which will be useful for other researchers.

Overall the MS is good and can be encouraged for publication.

Author Response

(The authors gave the same response as above.)

Reviewer 3 Report

Terpenoids are the largest group of natural products, which include both primary and specialized compounds. Although the biosynthetic pathway of terpenoids has largely been elucidated, their intracellular and intercellular transportation is poorly understood, for which the diverse characteristics and physiological activities are two major reasons.

In this manuscript, the authors provide a comprehensive review of current understandings of how terpenoids are transported.

I have only a few minor comments.

1. Line 18, "the thousand terpenoids identified" is not corrected. I believe that dozens of thousands of terpenoid structures have already been elucidated from all plants.

2. There are quite some grammar and typo issues in the manuscript. For example, in line 59, "Beside" should be "Besides", line 126, "such rice" should be "such as rice", line 133, "hormon" should be "hormone", etc. Please check carefully in the revision.

3. Line 165, carotenoids are light-harvesting pigment, but not "photoreceptor". The photoreceptor is a signaling component such as phytochrome.

4. Section 3.4.1. When talking about the transport of ABA, I am curious whether there has been some information on the transportation of xanthoxin from chloroplasts to the cytosol. It is incorrect to say that xanthoxin and ABA-aldehyde are both cytosolic intermediates (line 359). Although xanthoxin is catalyzed to synthesize ABA in the cytosol, itself is synthesized in chloroplasts.

5. I realize that there is limited information on the transport of monoterpenoids. I believe a key reference should be cited and discussed. In 2017, Adebesin et al. reported "Emission of volatile organic compounds from petunia flowers is facilitated by an ABC transporter" in Science (356: 1386-1388). This milestone work should not be missed.

Author Response

(The authors gave the same response as above.)
